

# Assessment of Indonesian Throughflow transports from ocean reanalyses with mooring-based observations

Magdalena Fritz[1,2], Michael Mayer[1, 2, 3], Leopold Haimberger[1], and Susanna Winkelbauer[1]

[1]Department of Meteorology and Geophysics, University of Vienna, Vienna, Austria
[2]b.geos, Korneuburg, Austria
[3]Research Department, European Centre for Medium-Range Weather Forecasts, Reading, UK

**Correspondence:** Magdalena Fritz (m.fritz@univie.ac.at)

**Abstract.** The transport of heat and freshwater from the Pacific to the Indian Ocean via the Indonesian Seas is commonly referred to as the Indonesian Throughflow (ITF). The interaction between the ITF and large scale phenomena occurring from intraseasonal to decadal time scales reflects its connection to the global ocean and the climate system, indicating the need for monitoring the ITF region. In situ observations in this region are highly valuable, but they are temporally and spatially insufficient for near real-time monitoring. Ocean reanalyses have the potential to serve as near-realtime monitoring tools but also to extended time series backward in time, yet a comprehensive quality assessment of their realism in this region with challenging bathymetry has been lacking so far. We focus on oceanic transports diagnosed from the Copernicus Marine Service (CMEMS) Global Reanalysis Ensemble Product (GREP) and the higher-resolution product GLORYS12V1, totalling six reanalysis products. They are validated against in situ observations taken from two different monitoring programs, namely International Nusantara Stratification and Transport (INSTANT 2004-2006) and Monitoring the Indonesian Throughflow (MITF 2006-2011 and 2013-2017), resulting in a total time series of about 11.5 years in the major inflow passage of Makassar Strait and shorter sampled time series in Lombok Strait, Ombai Strait, and Timor Passage. Results show that there is reasonable agreement between reanalysis-based transports and observations in terms of means, seasonal cycles, and variability, although some shortcomings stand out. The lower resolution products do not represent the spatial structure of the flow accurately. They also tend to underestimate the integrated net flow in the narrower straits of Lombok and Ombai, an aspect that is improved in GLORYS12V1. Reanalyses tend to underestimate the effect of seasonal Kelvin waves on the transports, which leads to errors in the mean seasonal cycle. Interannual variations of reanalysed transports agree well with observations, but uncertainties are much larger on sub-annual variability. Finally, as an application of physically consistent and observationally constrained fields provided by ocean reanalyses, we study the impact of the vertically varying pressure gradient on the vertical structure of the ITF to better understand an apparent two-layer regime of the flow.




# 1 Introduction

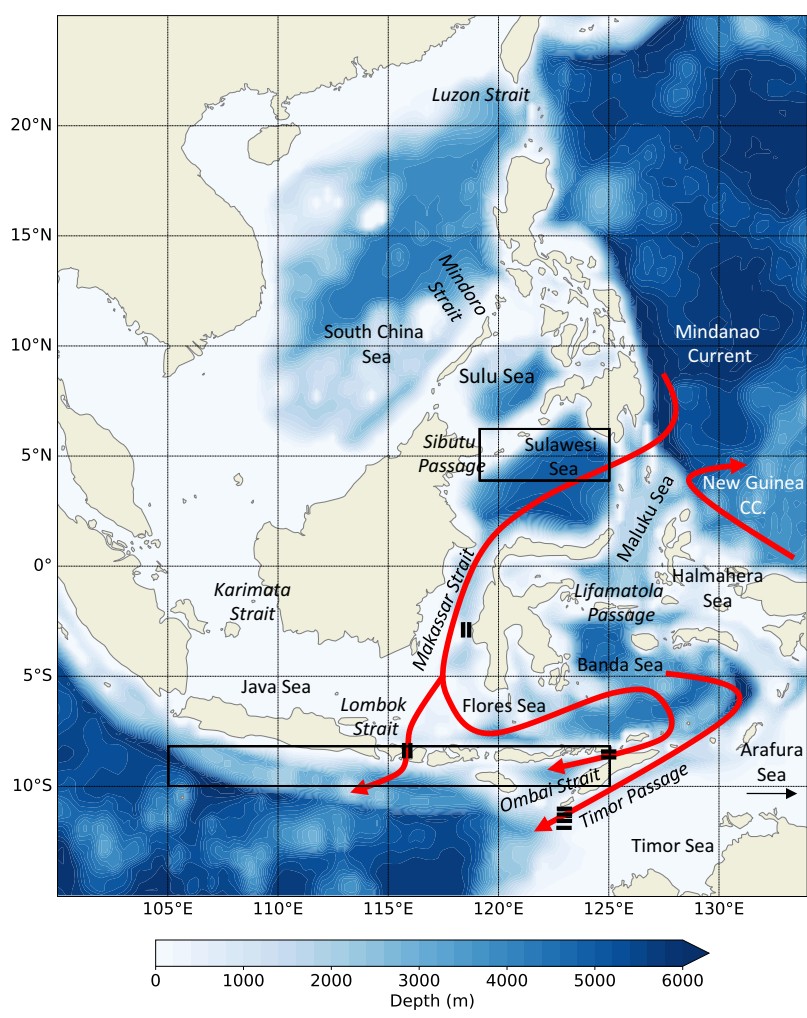

**Figure 1.** Study area: the Indonesian Seas. Solid and dashed red arrows display schematics of the Indonesian Throughflow and the South China Sea Throughflow, respectively. Bold black lines indicate mooring sites during the INSTANT program. Black encircled areas correspond to the ITF's entrance (119° E-125° E; 4° S-6° S) and exit (105° E-125° E; 8° S-10° S) regions.

The Indonesian Seas (Fig. 1) are the primary low latitude connection between the global oceans that allow the transport of heat and freshwater from the Pacific to the Indian Ocean (Piola and Gordon, 1984; Vranes et al., 2002; Potemra et al., 2003). This connection is known as the Indonesian Throughflow (ITF, Wyrtki (1961)). The Indonesian Archipelago is characterized by many narrow and deep straits connecting seas and basins of varying sizes and depths. The region is relevant because changes in sea surface temperature over the Indo-Pacific warm pool, where one ascending branch of the Walker Circulation lies, is strongly coupled to the atmosphere and hence can modulate atmospheric circulation in the global tropics and beyond (God-





frey, 1996; Sprintall et al., 2014). Two western boundary currents prevail at the entrance of the Indonesian Seas: the North
Pacific Mindanao Current (Schönau et al., 2015) and the South Pacific New Guinea Coastal Current (Cresswell, 2000). The

western boundary currents collide and form the ITF as well as the retroflections that feed the North Equatorial Counter Current
(Wyrtki and Kendall, 1967). Subsequently, North Pacific upper thermocline waters make their way through the Sulawesi Sea
into Makassar Strait (∼250 km), which accounts for about 80% of the volume inflow to the ITF (Gordon, 2005). In the west,
the flow through Makassar Strait is influenced by a large shelf restricting most of the transport to the 45 km wide Labani
Channel. After transiting Makassar Strait, water masses either enter the Banda Sea through the Flores Sea or directly exit into

the Indian Ocean via the shallow Lombok Strait (∼35 km) (Boy, 1995; Sprintall et al., 2009). We note that Lifamatola Passage
represents an alternative pathway to Makassar Strait, through which South Pacific water masses can enter the Banda Sea as well
(Van Riel, 1956; Van Aken et al., 2009) . From the Banda Sea, the ITF enters the Indian Ocean through small gateways along
the Nusa Tenggara island chain (Godfrey, 1996), but mainly through Ombai Strait (∼37 km) (Molcard et al., 2001; Sprintall
et al., 2009) and Timor Passage (∼160 km) (Molcard et al., 1996; Sprintall et al., 2009).


The ITF exhibits fluctuations on a broad range of time scales, from interannual time scales associated with the El Niño
Southern Oscillation (ENSO) (Mayer et al., 2018; Mayer and Alonso Balmaseda, 2021) or the Indian Ocean Dipole (IOD)
(Potemra and Schneider, 2007; Pujiana et al., 2019), to decadal climate variability and its connection to the Pacific Decadal
Oscillations (PDO) (Nieves et al., 2015; Ummenhofer et al., 2017). On shorter time scales, the Madden Julian Oscillation

(MJO) (Nieva Tamasiunas et al., 2021) and the Australian-Indonesian monsoon (Clarke and Liu, 1993; Masumoto and Ya-
magata, 1996) strongly impact the behaviour of the ITF. Related to the latter, the mean seasonal cycle of the ITF is dictated
by trade winds over the western Pacific and reversing wind patterns tied to the monsoon phases, as first postulated by Wyrtki
(1987). Together they maintain an inter-ocean pressure gradient between the western Pacific and the eastern Indian Ocean that
regulates the ITF.


During the southeast (SE) monsoon, southeasterly winds blow along the coast of Sumatra, Java, and the Nusa Tenggara is-
land chain, and this, as a result of Ekman transports (Ekman, 1905), pushes water masses offshore (Masumoto and Yamagata,
1996), resulting in a local mean sea level decrease. This, in turn, increases the inter-ocean pressure gradient towards the Indian
Ocean and favors a stronger southward ITF transport. During boreal winter, when the northwest (NW) monsoon prevails, the

opposite holds true. Accordingly, we can understand the seasonal cycle of ITF transport by studying the inter-ocean pressure
gradient between the ITF's entrance and exit region, which at $z = 0$ is proportional to the sea level gradient.

Several studies successfully employed ocean reanalyses to quantify different aspects of ocean climate, e.g., volume and heat
transport (Pietschnig et al., 2017), ocean heat content (Balmaseda et al., 2013; Palmer et al., 2017; Asbjørnsen et al., 2019;

Uotila et al., 2019), and energy budgets (Mayer et al., 2019, 2022).In the ITF area, ocean reanalyses have also been used to
study multidecadal (Ummenhofer et al., 2017) and interannual (Mayer et al., 2018) anomalies (related to, e.g., PDO, ENSO,
and IOD) of transports, which regulate the Indo-Pacific heat transfer. However, the ITF region is a challenging area for reanal-



ysis products given the complex bathymetry, and comprehensive validation in the ITF region is still lacking. Here we aim to fill
this gap by studying all relevant straits of the ITF (Makassar, Lombok, Ombai, and Timor) as represented by multiple ocean

reanalysis products and validation through available in situ observations. For this purpose, we employ in situ observations from
the aformentioned International Nusantara Stratification and Transport (INSTANT) program (Sprintall et al., 2004), providing
three years of data in the major inflow and outflow passages, and the Monitoring the Indonesian Throughflow (MITF) cam-
paign, providing temporally extended data for Makassaar Strait.

The rest of the paper is organized as follows: Section 2 introduces the data sets and the preprocessing methods. We con-
tinue with a comprehensive comparison between mooring observations and six reanalysis products using suitable diagnostics
in Sect. 3. Furthermore, we focus on the relation between the ITF and the vertically-varying pressure gradient. Conclusions
follow in Sect. 5.

## 2  Data and methods

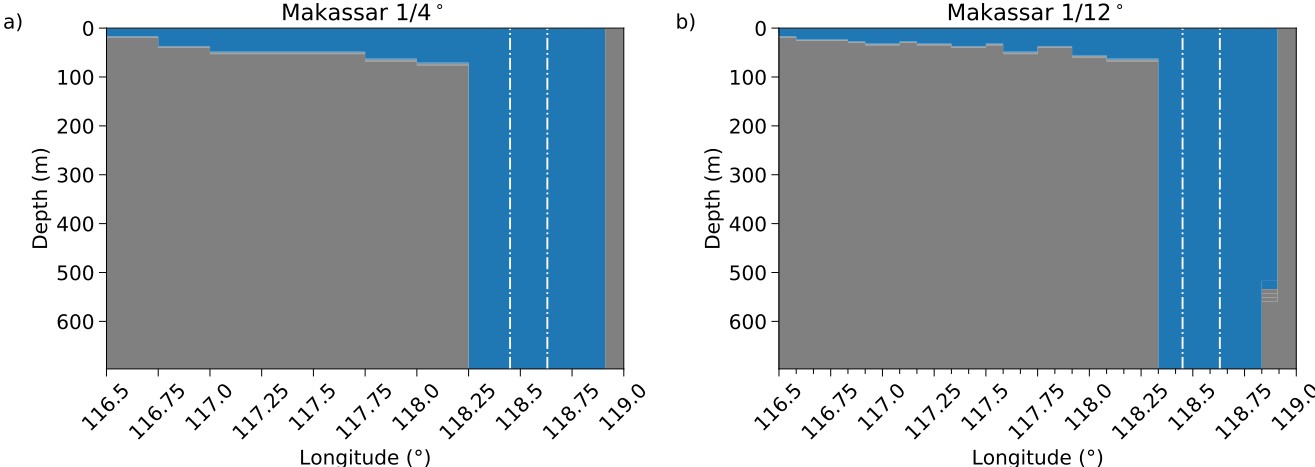

**Figure 2.** Bathymetries for the cross-passage interpolation of ASVs as given by the a) $1/4° \times 1/4°$ (75 levels) and b) $1/12° \times 1/12°$ (50 levels) products in Makassar Strait. White dash-dot lines represent mooring locations.

### 2.1  Mooring data

Observational data used throughout this study were measured during the INSTANT field program (August 2003 to December
2006) in Makassar Strait (Gordon et al., 2008), Lombok Strait, Ombai Strait, and Timor Passage (Sprintall et al., 2009). Mea-
surements continued in Makassar Strait from December 2006 to August 2011 and from August 2013 to August 2017 (MITF
(Gordon et al., 2019)), yielding a total of 11.5 years of observational data in Makassar Strait. The moorings were instrumented

with upward-looking Acoustic Doppler Current Profilers (ADCPs) and various current meters to measure zonal ($u$) and merid-



ional ($v$) velocities, as well as temperature, salinity, and pressure (Cowley et al., 2009). In order to obtain along-strait velocities, we employed the following preprocessing routine: first, we resample the data onto a common time base of 2 hours and deal with missing velocities, which mostly occurred in the surface layer, as follows. As in Sprintall et al. (2009), gaps were filled using linear interpolation or constant velocity equal to the shallowest measured velocity, i.e., nearest vertical neighbours. A data set

for each mooring is then created by linearly interpolating the observations to the vertical levels as defined in the reanalysis products. Measurements from ADCPs require special treatment to account for mooring blow-over. We compiled data sets for the $u$ and $v$ components (as measured by the current meters), and generated along strait velocities (ASVs) by projection of their contribution onto the along-strait vector $\boldsymbol{n}$.

According to Gordon et al. (2008) and Sprintall et al. (2009), the orientation of the along-strait vector $\boldsymbol{n}$ can be determined in several ways. We use a geometric approach, independent of any measurements (e.g., pressure gauges). Since the moorings within a strait are almost all aligned perpendicular to the strait, $\boldsymbol{n}$ is defined as the vector normal to the direction vector between the two outermost mooring locations. Accordingly, the along-strait velocity is the sum of the projections of $u$ and $v$ on $\boldsymbol{n}$. To estimate the transport through each strait, the velocity profiles at each mooring site are laterally interpolated (within 10

m bins) between each other and extrapolated to the sidewalls. The inter/extrapolation is performed within the bathymetry as specified by the 1/4° (Fig. 2a) and 1/12° (Fig. 2b) reanalysis products while assuming a linear drop-off to zero towards the shore. Integrating over the cross-passage interpolation of ASVs yields mean volume transports.

Significant gaps occur for the Timor Sill data from August 2004 to June 2005 and during the second deployment, where data

is only available between ∼300-1900 m. Since only the west mooring was operational during the MITF program (2006-2017), velocity data at the eastern site was estimated using a Linear Regression model, which was motivated by the high correlation of velocities measured at the east and west mooring during the INSTANT period. Also, observational data for the INSTANT period were preprocessed as mentioned above, while MITF data were downloaded, and fully preprocessed, without detailed documentation on the methods (our technique closely follows the processing described in (Gordon et al., 2008; Sprintall et al.,

2009)). We apply the same correlation-based approach to the Lombok west mooring, where data is unavailable after June 2005 due to an early mooring parting. Data output from individual instruments that exhibited significant gaps or stopped working entirely were generally not considered.





## 2.2 Reanalysis data

**Table 1.** Product name (institution), resolution (horizontal and vertical), ocean model (version), atmospheric forcing, and data assimilation method for the considered reanalyses.

| Product | Resolution | Ocean Model | Forcing | Data Assimilation |
|---|---|---|---|---|
| C-GLORS (v7) (CMCC) | ORCA025.L75 | NEMO (v3.6) | ERA-Interim | 3D-Var/FGAT (OceanVar[1]) |
| FOAM (Met Office) | ORCA025.L75 | NEMO (v3.2) | ERA-Interim | 3D-Var/FGAT (NEMOVAR[2]) |
| ORAS5 (ECMWF) | ORCA025.L75 | NEMO (v3.4) | ERA-Interim | 3D-Var/FGAT (NEMOVAR) |
| ORAP6 (ECMWF) | ORCA025.L75 | NEMO (v3.4) | ERA5 | 3D-Var/FGAT (NEMOVAR) |
| GLORYS2V4 (Mercator Ocean) | ORCA025.L75 | NEMO (v3.1) | ERA-Interim | Kalman Filter (SAM[3]) |
| GLORYS12V1 (Mercator Ocean) | ORCA12.L50 | NEMO (v3.1) | ERA-Interim | Kalman Filter (SAM) |

[1](Dobricic and Pinardi, 2008), [2](Mogensen et al., 2012), [3](Lellouche et al., 2018)

We evaluated the following reanalysis products: CMCC Global Ocean Physical Reanalysis System (CGLORS) (Storto and
Masina, 2016), Forecasting Ocean Assimilation Model (FOAM) (MacLachlan et al., 2015), Global Ocean Reanalysis and
Simulation Version 4 (GLORYS2V4) (Garric and Parent, 2017), Ocean Reanalysis System 5 (ORAS5) (Zuo et al., 2015), and
Ocean ReAnalysis Pilot system-6 (ORAP6) (Zuo et al., 2021). These products use the NEMO ocean model (Madec et al.,
2008) in the ORCA025.L75 (Madec and Imbard, 1996) configuration, indicating an eddy permitting horizontal resolution of
$1/4° \times 1/4°$ ($\sim$28 km near the equator) and 75 vertical levels. The first four products (excluding ORAP6) contribute to the
Copernicus Marine Service (CMEMS) Global Reanalysis Ensemble Product (GREP) (Desportes et al., 2017), however, in this
work, we also consider ORAP6 as a contributor to our (extended) GREP mean. Different versions of the NEMO model are
implemented as a tripolar ORCA grid with Arakawa C-grid staggering (Arakawa and Lamb, 1977), where individual variables
are computed at different grid points with scalars defined at the cell center and vector components defined at the cell edges.
The sixth considered product GLORYS12V1 (Lellouche et al., 2018) is also based on the NEMO model, but with a increased
horizontal resolution of $1/12° \times 1/12°$ ($\sim$9 km near the equator) and 50 vertical levels. Within the upper 200 m, the $1/4°$
products provide data on 31 unevenly distributed vertical levels, whereas in GLORYS12V1, there are 25 levels. The consid-
ered reanalysis products employ different data assimilation methods (Table 1), assimilating satellite observations of sea level
anomalies, in situ observations of sub-surface temperature and salinity, as well as remotely sensed sea ice concentration (SIC)



and sea surface temperature (SST). In situ observations are extracted from the quality controlled UK Met Office EN4 data sets
(Good et al., 2013) or the CORA data set from the Institut Français de Recherche pour l'Exploitation de la Mer (IFREMER)
(Cabanes et al., 2013), which include a huge collection of in situ profiles of temperature and salinity. We note that coverage by
Argo moorings (Argo, 2000) is limited in the ITF region, i.e., observational coverage is reduced in the ITF compared to other
tropical regions. Also note that current measurements from mooring buoys are not assimilated, which makes the intercompar-
ison with in situ observations of oceanic currents completely independent. The ECMWF ERA-Interim atmospheric reanalysis
(Dee et al., 2011) is used to force each of the reanalyses at the surface using CORE bulk formulas (Large and Yeager, 2004),
except ORAP6, which employs updated forcing based on ERA5 (Hersbach et al., 2020).

Since all products use Arakawa C-grid staggering, the definition of an ASV vector is not straightforward. The procedure
yielding cross sections of ASVs follows that of the observations, however, the projection from the native grid onto the ge-
ographic coordinates in the desired strait requires the implementation of three cross products and, again, a projection of the
direct velocity vectors $u$, $v$ onto the along strait vector $n$.

## 3  Comparison between observations and reanalyses

### 3.1  Vertical profiles

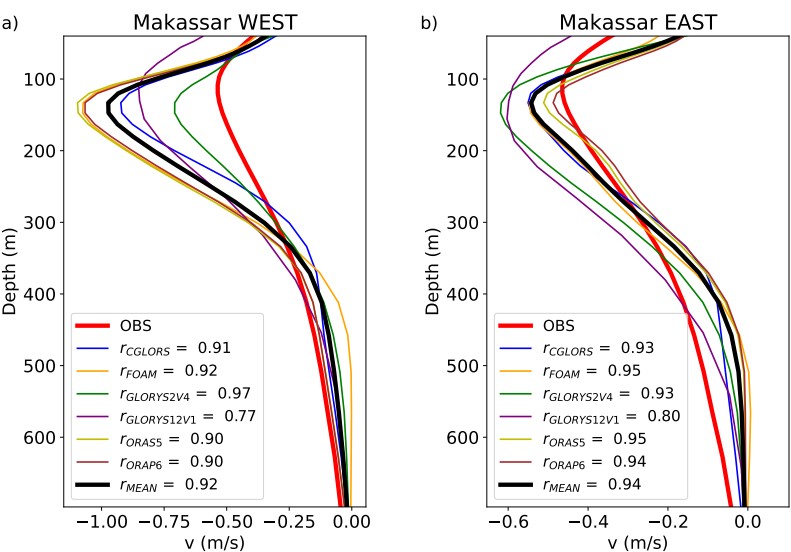

**Figure 3.** Mean ASV profiles for a) Makassar west and b) Makassar east. Corresponding correlation coefficients between observations and
reanalyses are given in the boxes in the lower left. Negative values indicate southward-directed velocities (towards the Indian Ocean).



We begin with an intercomparison of mean vertical profiles of ASVs in Makassar Strait (Fig. 3). Understanding differences
in the depiction of velocity profiles between the observations and reanalysis products is essential for the interpretation of dif-
ferences between them. The GREP profiles correspond to the nearest neighbour profiles of the west-and east mooring site.
GLORYS12V1 west coincides with the second nearest neighbour due to a shelf-induced artifact that apparently corrupts the
true nearest neighbour profile (not shown). Due to this artifact in GLORYS12V1, the longitude (118.5° E) of the chosen verti-
cal profile coincides with that of the GREP products.


Vertical profiles of ASV indicate stronger transport to the west of Labani Channel in observations, which is also evident in the
reanalysis products, however, greatly overestimated by up to -0.56 m s$^{-1}$. Observations display a mean maximum velocity of
-0.54 m s$^{-1}$ in ~110 m, while the GREP mean exhibits almost twice as high velocities with up to -0.97 m s$^{-1}$ in ~130 m. The
lower placed maximum is featured in all considered reanalysis products. The best agreement in terms of maximum currents
is found in GLORYS2V4 (vmax = -0.71 m s$^{-1}$). It also exhibits the highest correlation (r = 0.97) with the observed vertical
ASV profiles (as a measure of the realism of their vertical structure). The maximum velocity in GLORYS12V1 (-0.85 m s$^{-1}$)
is weaker than in the GREP mean and thus closer to observations but still noticeably overestimated. In addition, the correlation
of the mean ASV profiles is lower for GLORYS12V1 (r = 0.77) compared to the GREP products and substantially lower than
its lower resolution counterpart GLORYS2V4. We note that we disregard the top 40 m in the computation of the correlation
coefficients due to surface reflection contamination in the observations that might adulterate the correlation results (Sprintall
et al., 2009; Gordon et al., 2019). In depths greater than ~400 m, reanalysis products approach near-zero velocities faster than
indicated by the observations. The too weak flow in GREP in greater depths is even more pronounced in the eastern profiles
(Fig. 3b), where observations display substantial magnitudes down to ~500 m. Qualitatively, mean reanalysis-based vertical
profiles at the eastern mooring site do not exhibit such a striking deviation of peak flow from observations. However, correlation
coefficients [0.8; 0.95] are similar as for the western mooring because the vertical displacement (~20 m) of maximum velocity
in the reanalysis products remains, while the observed maximum amounts to -0.46 m s$^{-1}$ in ~110 m. The GREP mean displays
a maximum value of -0.54 m s$^{-1}$, surpassed by both GLORYS products (vmax$_{GLORYS2V4}$ = -0.62 m s$^{-1}$, vmax$_{GLORYS12V1}$ = -0.6
m s$^{-1}$). While ORAS5 and ORAP6 draw the GREP mean profile towards lower velocities, GLORYS2V4 and GLORYS12V1
act to increase its magnitude at the eastern mooring site, which is the other way around as at the western location. Velocities
from reanalyses below ~400 m decrease more strongly in all products than in observations, which influences further results as
well.





## 3.2 Cross sections (spatially)

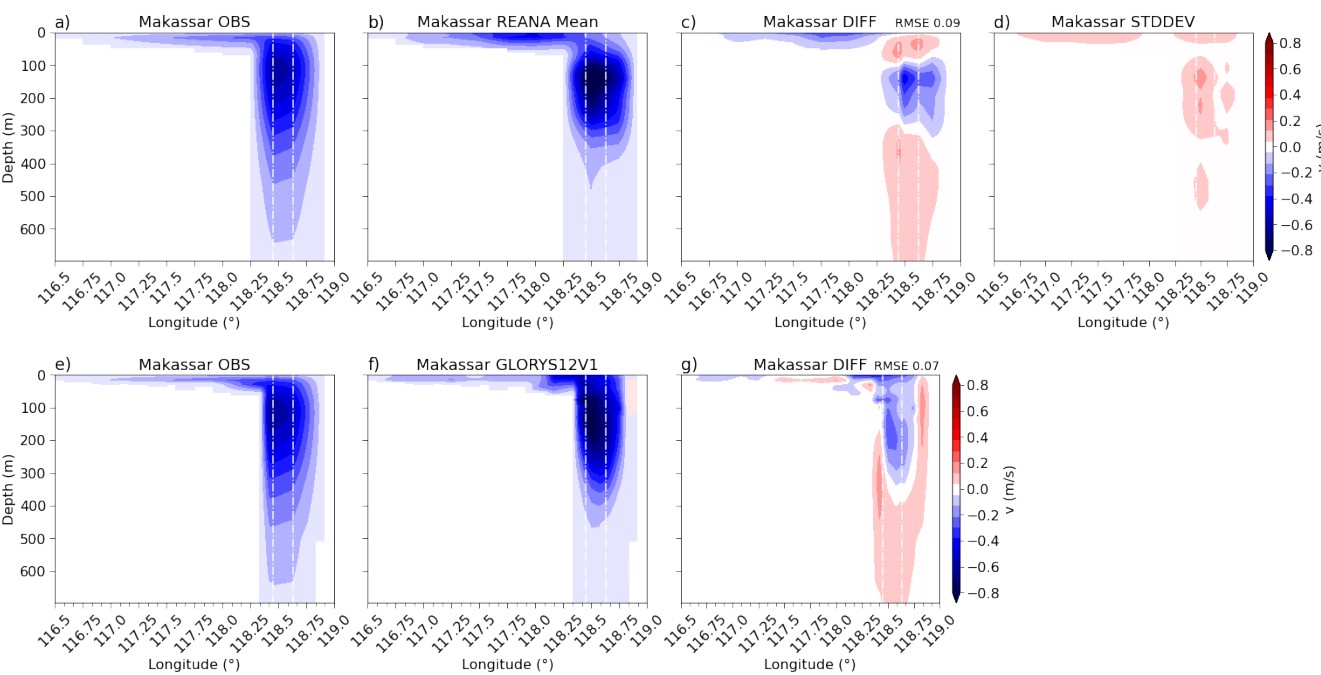

**Figure 4.** Mean ASV cross sections for Makassar Strait as given by a,e) the observations, b) the GREP mean, and f) GLORYS12V1 for the $1/4°$ (upper panel) and the $1/12°$ (lower panel) bathymetry. c,g) correspond to the respective differences between observations and GREP mean with the RMSE given in the top right corner. The standard deviation of the individual products around the GREP mean is shown in d). White dash-dot lines represent mooring locations. Negative values indicate southward-directed velocities (towards the Indian Ocean).

Figure 4 represents cross sections of temporally averaged (2004-2017) ASVs for Makassar Strait. It shows that the location of the stronger core of the western mooring in Makassar Strait is not as well resolved by the GREP mean (Fig. 4b) as in

GLORYS12V1 (Fig. 4f). This most likely corresponds to the increase in available horizontal grid points in GLORYS12V1 and hence to the fact that the flow exhibits a richer structure. The observed western intensification of the flow through the Labani Channel in Fig. 4a is consistent with findings by Gordon et al. (2008). In fact, we found cross-strait differences in the strength of the flow through most of the straits (Fig. A1): in Lombok Strait, the western side is stronger than the east, and in Ombai Strait, the southern side carries nearly the complete transport through this strait. The $1/4°$ products struggle to resolve such

asymmetries in the flow through the passages. The comparatively low RMS differences between the cross-sections from the individual reanalysis products and the GREP mean (not shown) justify the frequently used comparison between observations and the GREP mean. Moreover, the standard deviations of the individual products around the GREP mean (Fig. 4d) are lower (RMSE $\sim0.05$ m s$^{-1}$) than the differences between the GREP mean and the observations. As for the magnitude and depth of the strongest southward flow, reanalyses overestimate maximum velocities in Makassar Strait and shift the core downward and

too far east, as already indicated in Fig. 3. Similar differences occur in the other passages, where maximum velocities tend to be




overestimated in broader and underestimated in narrow straits, respectively (Table 2). The higher resolving bathymetry (Figs. 4e-g) leads to better agreement between observations and GLORYS12V1 as indicated by the RMSE in Fig. 4g. Furthermore, the difference plots in Figs. 4c and 4g emphasize that not only the 1/4° products but also GLORYS12V1 exhibit too weak currents in greater depths. Cross-sections for the other considered straits are provided as supplementary material (Fig. A1).

## 3.3 Long-term mean integrated transports

The cross-sections of ASVs form the basis for the computation of integrated transports. Cross-passage interpolation of ASVs within the GREP bathymetry (1/4°, Fig. 2a) yields integrated volume transports (2) of -9.9 Sv and -10.6 Sv for the observations and the GREP mean, respectively. In comparison, within the 1/12° bathymetry (Fig. 2b), volume transports amount to -9.1 Sv for observations and -9.2 Sv for GLORYS12V1. The interpolated transport over the western shelf amounts to -1.5 Sv in the GREP mean, but remains > -1 Sv in the observations and GLORYS12V1. Our observational results for both bathymetries are lower than the values obtained by Gordon et al. (2019) (-12.5 Sv between 2004 and 2017 and -11.6 Sv for the INSTANT period (Gordon et al., 2008)). It is however unclear which bathymetry was used in those studies. Also, experimentation with the cross-passage interpolation method revealed that results strongly depend not only on the choice of bathymetry but also on assumptions about boundary conditions. Hence, the difference between our observation-based results and those from Gordon et al. (2008) likely reflect uncertainties arising from the extrapolation of two measured profiles to a complete section across one strait.

**Table 2.** Mean integrated volume transports as given by the 1/4° (3rd, 4th, and 5th column) and 1/12° (6th, 7th, and 8th column) bathymetry. NN/INT refers to values where only the nearest neighbours (NN) or interpolated profiles (INT) were considered. Transport averages are given in Sverdrup (Sv). Negative values indicate southward-directed transport (towards the Indian Ocean).

| Strait | Time Period Average | OBS (1/4°) | GREP | GREP (NN/INT) | OBS (1/12°) | GLORYS12V1 | GLORYS12V1 (NN/INT) |
|---|---|---|---|---|---|---|---|
| Makassar | 2004-2017 | -9.9 | -10.6 | -9.7 | -9.1 | -9.2 | -10.8 |
| Lombok | 2004-2006 | -2.6 | -1.5 | -1.1 | -2.6 | -2.5 | -2.5 |
| Ombai | 2004-2006 | -5.3 | -3.5 | -4.1 | -5.1 | -4.0 | -3.4 |
| Timor | 2004-2006 | -6.7 | -8.9 | -9.7 | -5.9 | -8.7 | -9.7 |

Mean transports provided in Table 2 highlight the better agreement (smaller discrepancies $\Delta$) between the observations and GLORYS12V1 (compared to the GREP) brought forward by the 1/12° bathymetry, especially in Makassar ($\Delta_{1/12°}$ = 0.1 Sv compared to $\Delta_{1/4°}$ = 0.7 Sv) and Lombok Strait ($\Delta_{1/12°}$ = 0.1 Sv compared to $\Delta_{1/4°}$ = 1.1 Sv). Taking into consideration that observed transport through Makassar is three times as strong as through Lombok, GLORYS12V1 seems to be able to accurately reproduce stronger and weaker mean velocities in both narrow and broad straits, respectively. Less obvious but still present, we find higher agreement between the observations and GLORYS12V1 in Ombai Strait ($\Delta_{1/12°}$ = 1.1 Sv) as well. Interestingly, discrepancies between observations and reanalyses in Timor Passage seem to be smaller when considering the



$1/4°$ values ($\Delta_{1/4°}$ = 2.2 Sv). However, it is evident that differences between observations and reanalyses are relatively large for both bathymetries, but they seem to be emphasized for the higher-resolved bathymetry ($\Delta_{1/12°}$ = 2.8 Sv). Sprintall et al. (2009) discussed results from the three outflow passages and found mean integrated transport estimates of -2.6 Sv in Lombok Strait, -4.9 Sv in Ombai Strait, and -7.5 Sv in Timor Passage, which is in good agreement with our observation-based estimates.

By considering only the nearest neighbours (NN) or interpolated profiles (INT) in Figs. 4b and 4f (instead of all available grid points), we can quantify how much information is lost by only observing at two sites and if the choice of mooring location represents the throughflow well. In the case of INT, we interpolated the NN to the actual mooring site in order to avoid using a NN twice in the narrow straits. The results show that the agreement between observations and the GREP mean (NN/INT) improves considerably ($\Delta_{1/4°}$ = 0.2 Sv) in Makassar. The strongly pronounced maximum in the west in GLORYS12V1 leads to

an exaggeration of ITF transport ($\Delta_{1/12°}$ = 1.7 Sv), but it also shows that the strongest flow is indeed captured by the moorings. The additional information from the grid points in the east is thus required to keep the integrated transport inbound, a result of the biased horizontal structure of the currents in the reanalysis. Overall, the differences between considering all available grid points and using only the NN/INT values amount to ∼10% in all straits. Thus, we consider this to be an appropriate representation of the sampling error for the limited number of moorings. The fact that the agreements between observations and

GREP/GLORYS12V1 and the agreements between observations and GREP (NN/INT)/GLORYS12V1 (NN/INT) vary in all considered straits (Table 2), highlight biased structures in the flow of the reanalyses and the need for an accurate representation of asymmetries by the reanalyses.

### 3.4   Mean annual cycle

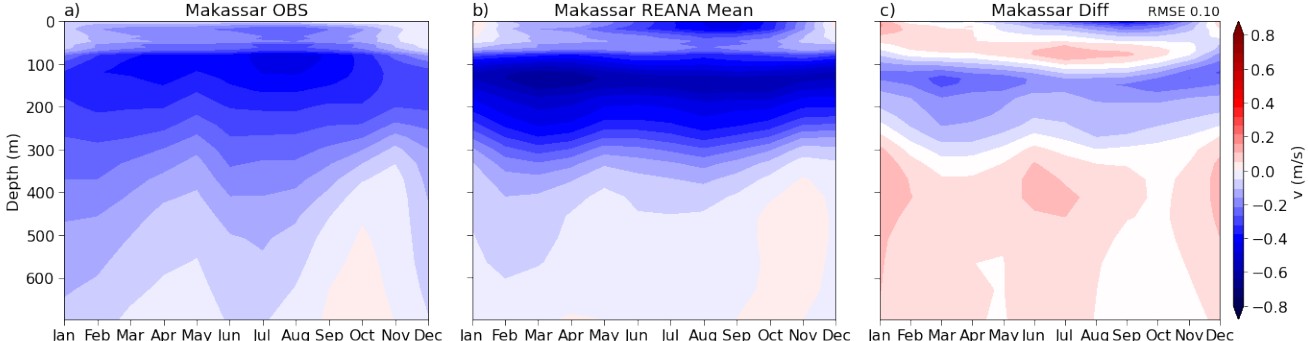

**Figure 5.** Mean annual cycle of ASVs for Makassar Strait as given by the $1/4°$ bathymetry for a) observations and b) the GREP mean. c) refers to their respective differences. RMSE between the observations and the GREP mean is given in the top right corner. Negative values indicate southward transport (toward the Indian Ocean).

The mean annual cycle of ITF transport strongly depends on the Australian-Indonesian monsoon and, thus, on seasonal vari-

abilities in the sea level differences between the western Pacific and the eastern Indian Ocean (Wyrtki, 1987; Clarke and Liu,





1994). Sea level differences, and hence transports, are at a maximum (higher in the western Pacific) in boreal summer during the SE monsoon, while minimum transports prevail during boreal winter (NW monsoon). Due to the differences in sea level, a pressure gradient between the Pacific and the Indian Ocean establishes, which ultimately drives the ITF. As will be discussed in section 4, vertically-varying pressure gradients regulate the throughflow all the way to the sea floor. Thus, ITF transport

through Makassar Strait exhibits a strong vertically-varying annual cycle as shown in Fig. 5a. The surface layer flow as well as the stronger subsurface flow in ∼100-120 m increase around July, and we find a second maximum around February/March, which is, however, less pronounced in the surface layer. The GREP mean (Fig. 5b) captures the seasonal cycle of the surface layer flow as well as the stronger subsurface core. In both layers, ASVs are considerably larger, explaining the negative differences (OBS – GREP mean) in Fig. 5c. The difference maximum around March highlights the overly high ASVs in the GREP

mean between February and mid-April.

During the monsoon transition months (April/May and October/November), the vertical structure in Makassar Strait changes, and weak ASVs (even positive) dominate lower layers. Such decreases in the throughflow represent the intrusion of Indian Ocean Kelvin waves that originate from anomalous wind forcing in the equatorial Indian Ocean and propagate along the south-

ern coast of Sumatra and Java into Makassar Strait via Lombok Strait (Sprintall et al., 2000), which act to reduce ITF transport by weakening the inter-ocean pressure gradient (Gordon et al., 2019). Hovmoeller plots indicate that Kelvin waves seem to appear in deeper layers first before influencing the ITF in the upper layer (Sprintall et al., 2009). Reanalyses generally have weaker flow in greater depths and the Kelvin wave signal appears suppressed in the ASVs from reanalyses. This has some influence also on the annual cycle as discussed next with respect to Fig. 6.





**Figure 6.** Mean annual cycle of ITF transport in a) Makassar Strait, b) Lombok Strait, c) Ombai Strait, and d) Timor Passage as represented by observations and reanalyses. Subjacent boxes display corresponding Pearson correlation coefficients r and RMSEs between the respective reanalysis and observations. Note that Makassar refers to the extended (2004-2017) observation period. Negative values indicate southward-directed transports (towards the Indian Ocean).

The condensed view of the mean annual cycle of vertically integrated transports in Figs. 6a-d) enables an intercomparison between all considered reanalyses and the observations. The mean annual cycle is most pronounced in Makassar Strait (Fig. 6a), exhibiting maximum transport of -12.5 Sv (July in OBS) and -13.6 Sv (August in GREP mean) and minimum transport of -5.4 Sv (November in OBS) and -6.6 Sv (December in GREP mean). The reduction in transport caused by Kelvin waves can also be assessed in the one-dimensional framework because transport minima in the monsoon transition months represent





the intrusion of Kelvin waves. Furthermore, the annual cycle in Makassar Strait reveals an apparent time lag of one month between the observations and all considered products, which is most pronounced in February/March and July/August. The lag arises both during the INSTANT period (3 years) and in the extended data set (January 2004 to August 2017). To exclude the possibility that the lag occurs just by chance, we assess the significance of the cross-correlation by computing significance barriers. The cross-correlation function peaks at lag 1 at r(1) = 0.78 (GREP mean), which is statistically significant on the 95%

level, taking autocorrelation in the individual series into account. This confirms a systematic difference between observations and reanalyses which will be addressed further below.

Lombok Strait (Fig. 6b), Ombai Strait (Fig. 6c), and Timor Passage (Fig. 6d) do not display such a lag. Also, these straits were only observed during the INSTANT program, meaning that we have a maximum of three years worth of data to deter-

mine the mean annual cycle. Representing the ITF's direct outflow passage, the seasonal cycle in Lombok Strait follows that of Makassar Strait (Fig. 6a): maximum transport during boreal summer (August in OBS -3.9 Sv; GLORYS12V1 -3.8 Sv) and minimum transport in boreal winter (December in OBS -1.1 Sv; GLORYS12V1 -0.7 Sv). In Lombok the observations display much stronger transports and also a stronger seasonal cycle compared to the GREP mean (and especially GLORYS2V4, ORAS5, and ORAP6), whereas GLORYS12V1 agrees better with the observations (based on the RMSE of the seasonal cycle

curve). This is likely due to its higher resolution and hence its ability to capture seasonal variabilities in narrow straits more accurately.

The annual cycle in Ombai Strait features distinct minima during the monsoon transition months in April (OBS -3.4 Sv) and October (OBS -3.2 Sv). According to Sprintall et al. (2009), some of the Kelvin wave energy propagating along Sumatra

travels further into Ombai Strait, explaining the minima in April and October. We find maximum transport during the NW monsoon (January in OBS -9.3 Sv) when both mooring profiles (not shown) display strong subsurface maxima. During the SE monsoon, Ombai transport increases again (July in OBS -6.5 Sv), however, it remains weaker than during the winter months. The GREP mean and GLORYS12V1 converge with the observations during the weak monsoon transition months and increase correctly in the main transport periods, January (GREP mean -4.5 Sv; GLORYS12V1 -5.6 Sv) and July (GREP mean -4.0 Sv;

GLORYS12V1 -4.9 Sv). Also here reanalysis-based transport strongly underestimate observed transport by up to ∼6 Sv in January (especially GLORYS2V4, ORAS5, and ORAP6). Furthermore, the 1/4° products (in contrast to GLORYS12V1) do not capture the asymmetry (positive velocities in the north) in the flow through Ombai Strait (Fig. A1d-f) further misrepresenting properties in the annual cycle.

The GREP products exhibit a large spread in Timor Passage, and their ensemble mean strongly deviates from the observations, not only in terms of magnitude but also in seasonal evolution. The observations exhibit minimum and maximum transport during August and September (Min in Aug -5.2 Sv) and between March and May (Max in Apr -7.9 Sv), respectively. Sprintall et al. (2009) have partly attributed minimum transports to Kelvin wave-induced deep sea flow reversals that are not captured by the 1/4° products, because they cannot represent flow below ∼1200 m due to their too shallow bathymetry. The GREP



mean displays an almost reversed cycle with minimum transport from February to April (Min in March -7.5 Sv) and maximum transport between July and September (Max in September -9.6 Sv). The seasonal cycle in GLORYS12V1 is particularly weak, ranging from -7.9 Sv in October to -9.2 Sv in March. However, the correlation between observations and GLORYS12V1 ($r_{GLORYS12V1}$ = 0.82) is by far the highest, keeping in mind the considerable differences in magnitude. Sprintall et al. (2009) suggest that when transport through Ombai Strait is at a minimum during the monsoon transition months (Apr/May and Oc-

t/Nov), Timor transport increases and vice versa as a result of the well known Wyrtki Jet (Wyrkti, 1973). Consequently, Kelvin waves associated with the Wyrtki Jet partly control the ITF's annual cycle (Sprintall et al., 2009) and therefore reanalyses need to be able to reproduce them.

From Fig. 6 it is evident that the sum of the outflow passages (Lombok, Ombai, and Timor) is stronger than the inflow

from Makassar Strait. This discrepancy is mostly explained by the flows through Lifamatola Passage. Measurements (single mooring) were taken in Lifamatola Passage during INSTANT, but they only cover depths greater than ∼400 m. Figure A2a and Fig. A2b show the total transport through the inflow (with Lifamatola contributions added) and outflow passages, respectively. The additional contribution from Lifamatola Passage shifts the maximum transport during the summer months from August to June in the reanalyses and from July to June in the observations, eliminating the one-month lag between reanalyses and

observations (except GLORYS12V1) we found in Makassar. Note that the reanalysis maximum does not shift when adding only the layer consistent with observations (∼400-950 m), suggesting a biased distribution of the reanalysed flow also in this strait. Furthermore, reanalysis-based results suggest that a considerable amount of information is lost by only observing in Lifamatola Passage without taking into account the whole eastern inflow route. Discrepancies between inflow and outflow transports (Fig. A2) increase around September and last until January, suggesting a more balanced ratio during the summer

months. The missing data in the upper 400 m and the fact that Lifamatola Passage covers only part of the eastern route likely explains those discrepancies.



## 3.5 Temporal variability

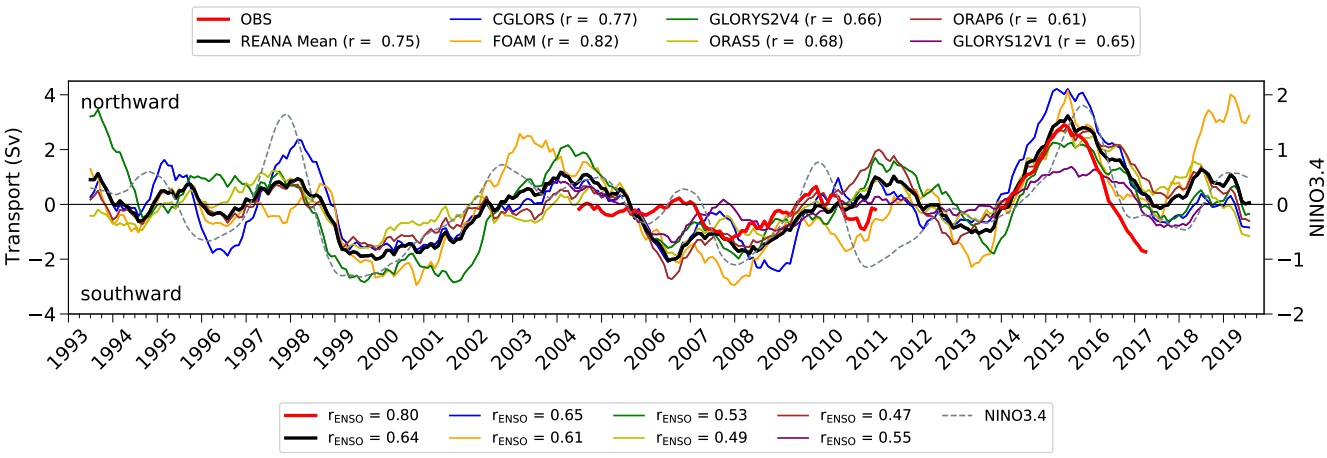

**Figure 7.** Monthly time series (with 12-month Moving Average) of total ITF transport anomalies through Makassar Strait as represented by the observations and the reanalysis products with NINO3.4 anomalies (Trenberth, 1997). Note the missing observational data between July 2011 and August 2013. Upper legend displays correlation coefficients r at lag = 0. Correlation coefficients $r_{ENSO}$ between transport anomalies and the NINO3.4 anomalies at lag = 0 are given in the lower legend. Negative values indicate southward-directed transports (towards the Indian Ocean).

Deseasonalized anomalies of Makassar volume transport between 1993 and 2019 are presented in Fig. 7. Here, positive (negative) anomalies refer to northward (southward)-directed transport anomalies. We find considerable spread between the different
reanalysis products, but anomaly correlations with observations are relatively high ($r_{lag=0}$ ranging between 0.61 (ORAP6) and 0.82 (FOAM)). Temporal standard deviations (STDDEV) of volume transport exceed 1 Sv in all time series (except ORAS5 0.98 Sv and GLORYS12V1 0.65 Sv), with the highest value in FOAM (1.59 Sv) and moderate deviations in the observations (1.05 Sv), underlining the high variability in the data sets. The signal-to-noise ratio (ratio between STDDEV of the GREP mean and spread of the GREP products (Balmaseda et al., 2015)) of 1.83 suggests that the products show reasonable agreement
in terms of interannual transport variability. NINO3.4 anomalies in Fig. 7 reveal a strong connection between ITF transport anomalies and ENSO (England and Huang, 2005). The time series show that the ITF is generally out of phase with ENSO and we find zero lag correlation coefficients between 0.47 (ORAP6) and 0.8 (OBS). The maximum lagged correlation coefficient reaches 0.86 (OBS) with transport lagging the NINO3.4 index by 3 months (not shown), suggesting that variabilities in the ITF are a lagged response to ENSO, as also found by England and Huang (2005).


The most prominent signal in the time series marks the period between 2014 to 2016, during which the El Niño event of 2015/16 occurred, when ITF transport in Makassar is usually reduced due to an eastward-directed anomalous pressure gradient in the Sulawesi Sea (Gordon et al., 2012). The effect of this strong ENSO event is visible in all products, albeit with varying





strength. The strong El Niño event is characterized by an extended period of substantial northward-directed anomalies, domi-
nated by transports in CGLORS. This corroborates results by Mayer et al. (2018) who, based on two reanalyses, found that the
strongest anomalies of ITF volume transport since 1993 were registered during this period. While observed transports become
negative again in mid-2016 with the onset of La Niña, there seems to be an approximately half-year delay in all reanalyses.

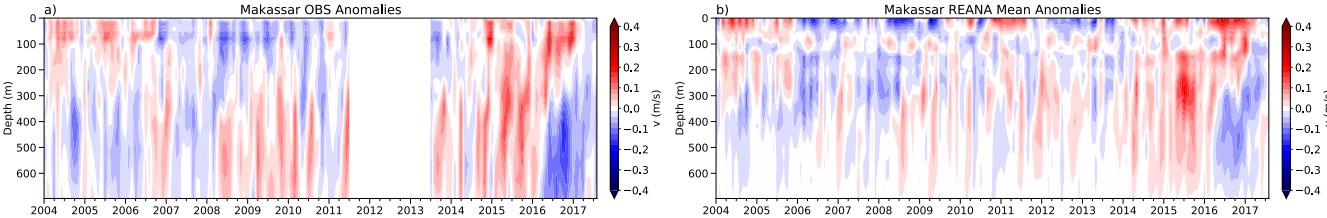

**Figure 8.** Monthly time series of ASV anomalies in Makassar Strait as represented by a) the observations and b) the GREP mean. Note the
missing observational data between July 2011 and August 2013. Negative values indicate southward-directed velocities (towards the Indian
Ocean).

We now turn to the cross-sectional view of monthly-averaged ASV anomalies (Fig. 8) to address the complex vertical struc-
ture of ITF variability in Makassar Strait. ASV anomalies in Fig. 8 highlight the well-matching structure between observations
and the GREP mean, especially in the upper 300 m. Focusing again on the strong El Niño event 2015/16, both observations
and reanalyses display intensified northward-directed transport anomalies (i.e. weakened flow) in the upper layer (<300 m)
and negative anomalies (i.e. strengthened flow) in the lower layer (>300 m). Negative anomalies represented by the GREP
mean are weaker and extend less far downward than in observations. Thus, the ENSO signal in integrated transports results
from compensating anomalies in the upper and lower layer, the balance of which varies across the different products and leads
to the considerable spread found in Fig. 7. We will address the effect of such compensating anomalies in detail in Sect. 4. The
La Niña event during 2007/08 marks another distinctive period of negative anomalies reaching down to ∼700 m, but again, the
GREP mean lacks a signal in deeper layers starting in around 300 m.




## 3.6 Performance summary metrics

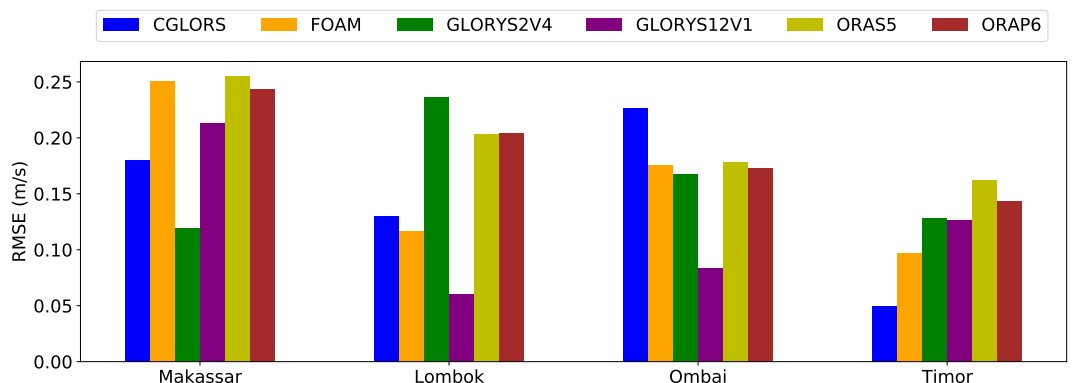

**Figure 9.** Reanalysis performance: RMSEs of time-averaged vertical profiles of ASV (in m/s) of reanalyses and observations in each strait. Note that Makassar covers the period from 2004 to 2017 while the other straits have data only from 2004 to 2006.

Based on the mean vertical profiles introduced in Fig. 3 and the integrated cross sections in Fig. 4 we now present a more systematic evaluation of all considered reanalyses in each strait. The evaluation rests on different applications of the root mean square error (RMSE) in reference to the observations and the bias of reanalyses. While the observations are certainly not free of errors, we regard them as truth here, any deviation from observations is considered an error.

Figure 9 shows the performance of each product in terms of RMSEs of time-averaged vertical profiles (NN/INT) of ASV in all straits. Results depend on the reanalyses' capabilities to accurately represent velocity profiles at the mooring sites. This, in turn, depends on the representation of asymmetries in the flow. There is considerable spread between the GREP products in Makassar Strait, with the RMSE of FOAM, ORAS5, and ORAP6 being almost twice as large as that of GLORYS2V4 (Fig. 3). The successor of ORAS5, ORAP6, shows a slight improvement. Although GLORYS12V1 is able to resolve grid points that are much more closely located to the mooring sites, its performance is surprisingly mediocre in Makassar. GLORYS12V1,
the higher resolving version of GLORYS2V4, exhibits a larger RMSE than GLORYS2V4. This can partly be attributed to the greatly overestimated flow in the surface layer in GLORYS12V1 (Fig. 3). We note that velocities are generally strongest in Makassar Strait, and hence the comparatively large RMSEs in this strait do not necessarily mean larger relative errors.

In Lombok Strait and Timor Passage, spread across the GREP products is also large, while there is a better agreement in Ombai Strait (within the reanalyses). GLORYS12V1 has an advantage over the 1/4° products particularly in the narrow Lombok and Ombai straits. In Lombok Strait, the stronger surface layer currents, especially towards the west, are accurately represented by GLORYS12V1, unlike the GREP products. Also, in Ombai Strait, opposite transport orientation in the upper ~200 m is only represented in GLORYS12V1 (not shown). There are four mooring sites in Timor Passage, and hence four comparisons
to consider (not shown). Discrepancies between observations and CGLORS are comparatively small in each layer at all sites,





which explains the good performance of CGLORS. Most 1/4° products yield better agreement away from the coast, whereas GLORYS12V1 seems to increase its skill towards the mainland (Timor; not shown).

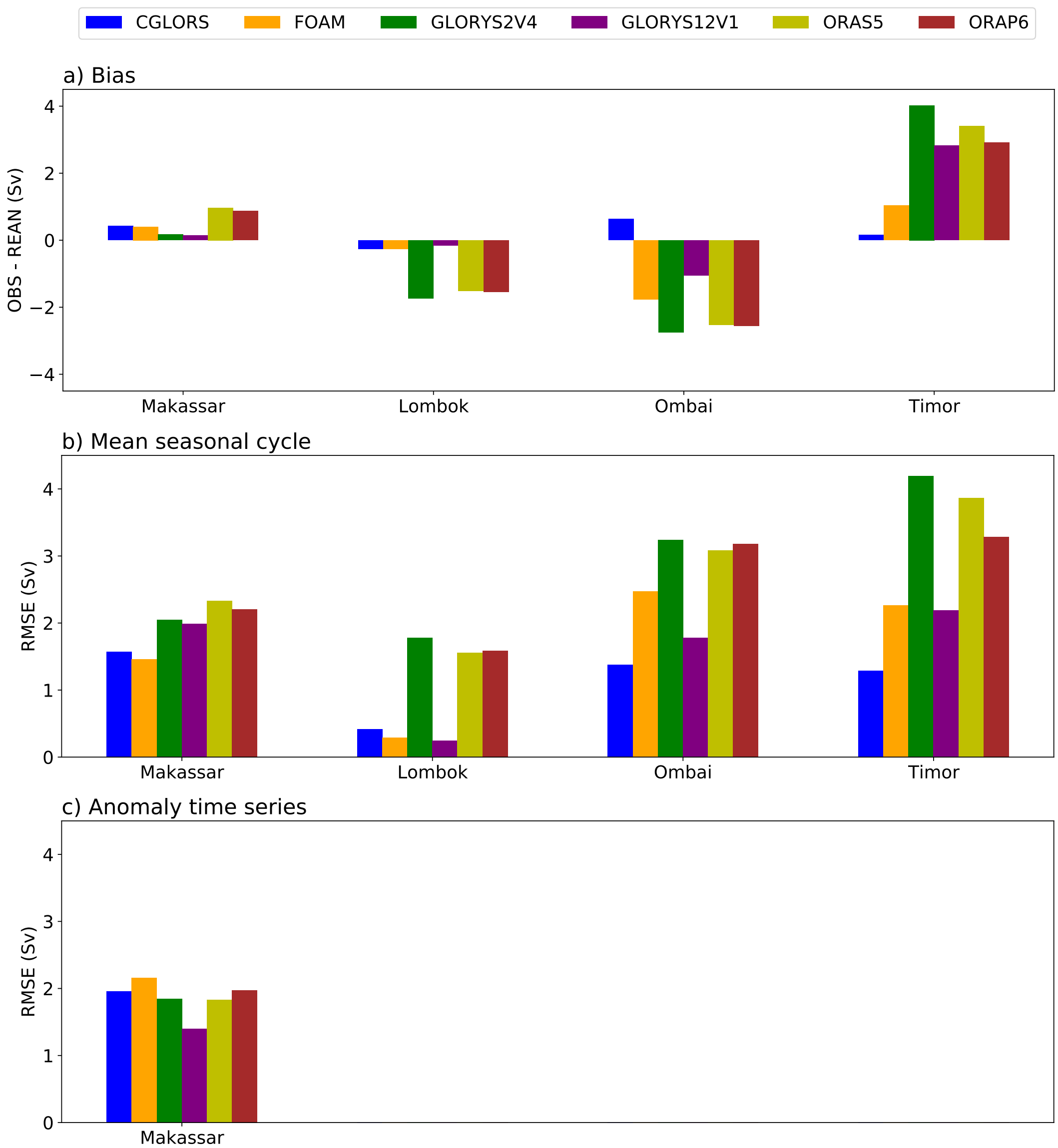

**Figure 10.** Reanalysis performance: a) Biases (in Sv) in reanalyses based on monthly time series of ITF transport in each strait. RMSEs of b) mean seasonal cycles of ITF transport (in Sv) and c) monthly time series of ITF transport (in Sv) anomalies and observations in Makassar Strait (outflow straits cover only 3 years and are therefore not considered). Note that Makassar covers the period from 2004 to 2017.

Figure 10 summarizes the performance of all reanalysis products based on monthly time series of integrated volume transport in each strait. Taking into account all horizontal grid points, these results reveal additional strengths and weaknesses of the





products in terms of performance. In Makassar Strait, the spread between the GREP products decreases considerably compared to Fig. 9. The performances of ORAS5 and ORAP6 are dominated by their overestimated velocities as represented by the bias (Fig. 10a) and the mean seasonal cycle (Fig. 10b). The bias points towards a moderate overestimation in all products, which is particularly small in the GLORYS products. By removing the seasonal pattern (Fig. 10c), the spread between the products barely changes, however, ORAS5 and ORAP6 no longer perform worst, and GLORYS12V1 displays its advantages.


Figure 6b already highlighted the considerable underestimation of transports in GLORYS2V4, ORAS5, and ORAP6 in Lombok Strait. Accordingly, these products exhibit a strong negative bias (Fig. 10a) and Fig. 10b confirms earlier results with the highest RMSE in GLORYS2V4. Two horizontal grid points are clearly not enough to cover transport through Lombok, still, CGLORS and FOAM exhibit good agreement with observations. However, this is not because they represent the asymmetry in

the flow correctly but because they provide generally high transport values (Fig. 6b), decreasing the RMSEs between observations and reanalyses. Thus, GLORYS12V1 (representing Lombok with four grid points) represents the spatial distribution of ASVs (not shown) correctly, likely explaining its low bias.

The GREP products cover also Ombai Strait with only two grid points, but the observed and reanalysed mean transport doubles

in magnitude (Table 2). Also here, the three reanalyses GLORYS2V4, ORAS5, and ORAP6 display considerable shortcomings in terms of the mean and seasonal variations (Fig. 6a and Fig. 6b). CGLORS shows good performance in Ombai (even outperforming GLORYS12V1). It exhibits strongest mean flow and the most accurate representation of the annual cycle.

Timor Passage displays high bias-corrected RMSEs, which is already evident from the inconsistencies we addressed in Fig.

6d. We find a similar ranking of the products as in Lombok and Ombai with a considerable discrepancy between the worst-performing (GLORYS2V4) and the best-performing product (CGLORS). GLORYS2V4 has a strong positive ASV bias towards the surface (not shown), which also appears in GLORYS12V1 (not shown) but is reduced towards the south due to additional grid points. Figure 9 already revealed the superior performance of CGLORS, which is reflected by the bias and the RMSE of the mean seasonal cycle likely caused by the non-overestimation of the flow in the upper ∼200 m (not shown; followed by

FOAM).



# 4 Two-layer regime

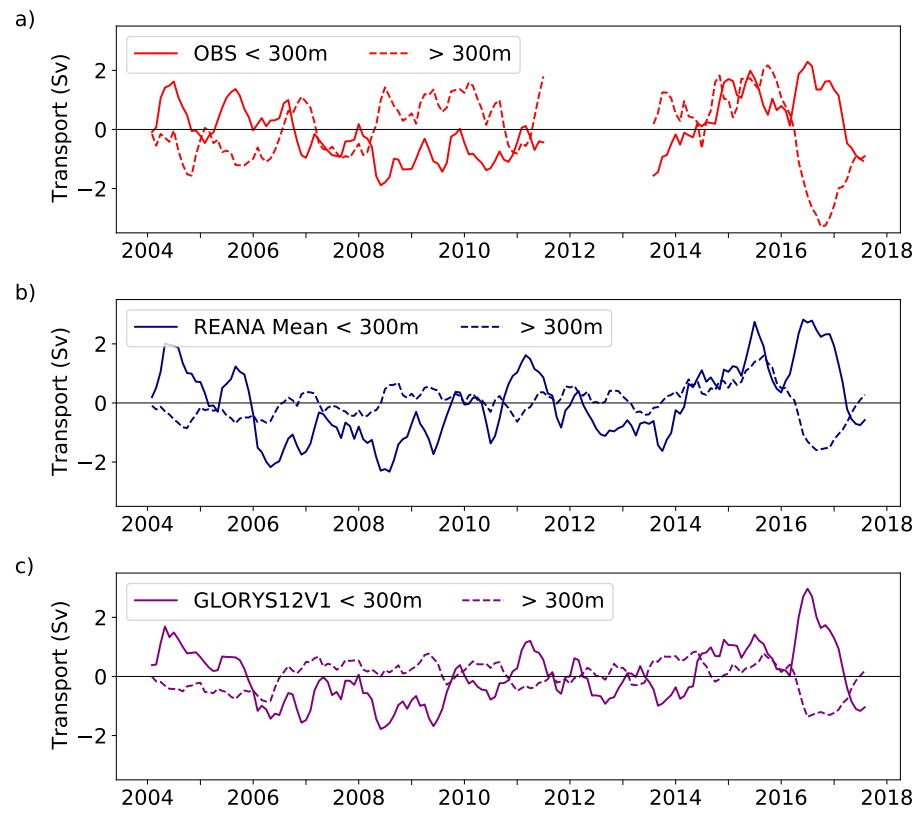

**Figure 11.** Monthly time series of ITF transport anomalies through Makassar Strait in the upper (continuous lines) and lower layer (dashed lines) as represented by a) observations, b) GREP mean, and c) GLORYS12V1. Considered time series cover the period between 2004 and 2017. Negative values indicate southward-directed transports (towards the Indian Ocean).

Dividing the water column in Makassar Strait into two layers, the upper layer ($<300$ m) and lower layer ($>300$ m) (Gordon et al., 2019; Pujiana et al., 2019), reveals interesting properties of the ITF. Figure 11 shows transport anomalies in the upper and lower layer as represented by the observations (Fig. 11a), the GREP mean (Fig. 11b), and GLORYS12V1 (Fig. 11c). There is a clear anti-correlation between upper and lower layers, with correlations of $r_{lag=-1}$ = -0.37, $r_{lag=-1}$ = -0.30, and $r_{lag=-1}$ = -0.45 respectively. By comparing Fig. 7 and Fig. 11, we find similar behaviour of full-depth integrated transport anomalies and upper layer anomalies ($r_{OBS}$ = 0.49), indicating that transport anomalies within the upper 300 m dominate the sign of the integrated anomaly. This behaviour is reproduced by the GREP mean ($r_{RMEAN}$ = 0.89) and GLORYS12V1 ($r_{GLORYS12V1}$ = 0.85). Observations display almost equally strong upper- and lower-layer anomalies, with even a dominant lower-layer anomaly in October 2016, after the strong El Niño event. The GREP mean and GLORYS12V1 exhibit comparatively weak lower layer anomalies ranging between [-1.6; 1.6] Sv and [-1.4; 0.9] Sv, respectively, which is consistent with their weak mean flow at depth (Fig. 3).



Monthly transport anomalies and monthly sea level gradient anomalies are negatively correlated in the upper layer ($r_{OBS}$ = -0.61, $r_{RMEAN}$ = -0.83, and $r_{GLORYS12V1}$ = -0.81). Accordingly, higher mean sea level anomalies in the Pacific (Indian Ocean)
correspond to southward-directed (northward) transport anomalies, which is consistent with the large-scale sea level gradient. The correlation reverses in the lower layer, indicating that the relation between transport and sea level difference anomalies no longer follows the force of the sea level gradient. Therefore, we conclude that the upper layer is regulated by the sea level gradient, while baroclinic effects dominate in lower layers.

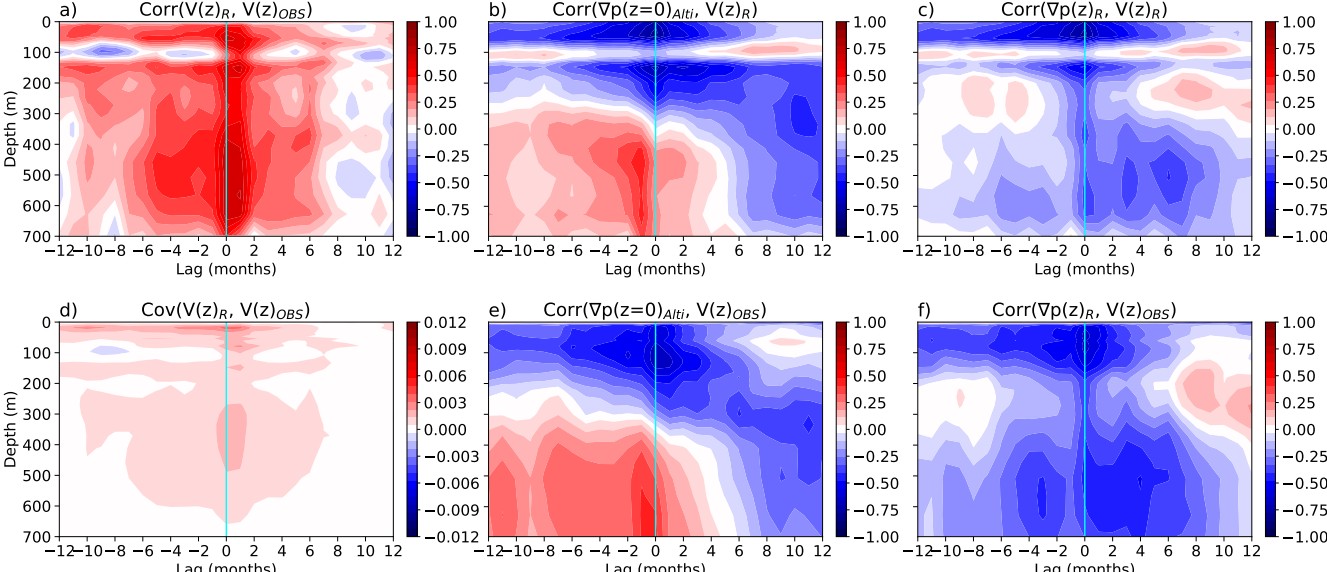

**Figure 12.** Depth-dependent lead-lag a) correlations and d) covariances between observed V(z)$_{OBS}$ and GREP mean based V(z)$_R$ ASVs anomalies in Makassar. Depth-dependent lead-lag correlations between monthly time series of b) GREP mean based sea level gradient anomalies $\nabla p(z = 0)_{Alti}$ and ASVs anomalies V(z)$_R$, and e) observed ASVs anomalies V(z)$_{OBS}$ in Makassar. Depth-dependent lead-lag correlation between monthly time series of GREP mean based pressure gradient anomalies $\nabla p(z)_R$ and ASVs anomalies c) V(z)$_R$, f) V(z)$_{OBS}$ in Makassar. Sea level gradients/pressure gradient anomalies refer to the ITF's entrance and exit region as indicated in Fig. 1. Considered time series cover the period between 2004 and 2017.

To investigate the apparent two-layer behaviour further, Fig. 12 presents diagnostics targeted at a better understanding of
depth-dependent relationships between oceanic flow and pressure gradients. First, Fig. 12a shows depth-dependent cross correlation of ASVs through Makassar Strait from observations and the GREP mean. We find maximum positive correlations at lag one month (from 70-200 m and 450-500 m), indicating that the GREP mean leads transport by about one month (Fig. 6a). A partition of the mean seasonal cycle into the upper (<300 m) and the lower (>300 m) layer reveals that the lag is more pronounced in the lower layer, where reanalyses exhibit weak current velocities. The maximum correlation at lag = 1 is also
present when performing a similar diagnostic only on the annual cycle of transports (Fig. A3a), which is consistent with the





results shown in Fig. 6a. We amplify its effect by considering a 10-year time series consisting only of the mean seasonal cycle.

Next, we study the relationship between the interocean pressure gradient and ASV anomalies in Makassar. The regions used for computation of the pressure gradient were chosen based on spatial correlation patterns between SLA and ASV anomalies (boxes in Fig. 1). Note that the chosen regions are similar to those used in Mayer et al. (2018) and Pujiana et al. (2019). Cross-correlations between pressure gradient anomalies at $z = 0$, which are proportional to the SLA gradient, and ASV anomalies for the GREP mean and observations are assessed in Fig. 12b and Fig. 12e, respectively. The results for both data types display similar properties: while maximum negative correlations ($r_{GREP}$ = -0.81, $r_{OBS}$ = -0.69) dominate the upper layer around lag 0, positive correlations govern the lower layer and become most notable towards lag -1. This confirms that sea level-gradient driven transports are an upper-layer phenomenon, and the lower layer is detached from the geostrophic flow near the surface. Covariances in Fig. 12d complement these results and highlight that the covariance between observations and reanalyses is still considerable in depths of ∼300-500 m.

Figs. 12c and 12f show cross correlations between the pressure gradient as a function of depth (based on reanalysis data) and ASVs from reanalyses and observations, respectively. The continuous negative correlations highlight that although transport anomalies in the upper and lower layer oppose each other, they are still regulated by the vertically-varying pressure gradient. ASVs and pressure gradients both decrease with depth, and accordingly, correlations decrease with depth (more so in Fig. 12c). Both observations and the GREP mean exhibit maximum correlation with the gradient at lag 0, indicating that the differences in Fig. 12a and Fig. A3a could arise from an "ageostrophic" part of the flow. Such processes have been attributed to baroclinic Kelvin waves that originate in the equatorial Indian Ocean and dictate the two-layer system of ITF transport anomalies (Pujiana et al., 2019). Drushka et al. (2010) traced the pathways of Kelvin waves from their point of origin (equatorial Indian Ocean) into the ITF region using altimetric SLA data and found that Kelvin waves are not linked to the large-scale sea level gradient but rather local anomalies, explaining the disappearing connection between sea level gradient anomalies and ASV anomalies in greater depths.

We note that correlations in the lower layer are higher for the observations. To assess the robustness of the reanalysis based pressure gradient, we attempted to obtain an observation based estimate of the pressure gradient using subsurface ocean temperature and salinity profiles from the EN4 data sets (UK Met Office (Good et al., 2013)), but insufficient data coverage did not allow to obtain robust results. Regarding the weak correlation around 100 m in Figs. 12b and 12c, we did verify that this is not due to cross-passage extrapolation over the shelf.

## 5 Conclusions

In this paper, we compared observations of oceanic flow in the ITF region from instruments installed on moorings with ocean reanalysis data. Firstly, the skill and limit of different ocean reanalysis products within Makassar Strait, Lombok Strait, Om-





bai Strait, and Timor Passage were assessed. This required careful preprocessing of both data types. Results are sensitive to
how the bathymetry was used for the cross-passage extrapolation of along strait velocities (ASVs) and the use of boundary
conditions, which indicates considerable uncertainties associated with transport calculations based on sparse mooring-based
measurements.

Intercomparisons between observations and reanalyses revealed generally positive (overestimation) and negative (underes-
timation) biases of transport in broad and narrow straits, respectively. GLORYS12V1 exhibited advantages in Makassar and
Lombok Strait, suggesting that the higher resolving reanalysis (more horizontal grid points) is capable of improving represen-
tations in both broader and narrow straits. Based on our diagnostics, we found that all reanalyses exhibit too weak currents
in greater depths. Furthermore, the 1/4° reanalysis products struggled with locating the core of the flow in the cross sections,
misrepresenting frequently occurring spatial asymmetries in the flow through the passages. As a result of the higher resolu-
tion of GLORYS12V1, the representation of such asymmetries did improve. This was also reflected in the long-term mean
integrated transports, where GLORYS12V1 agrees much better with observations than the other GREP products in all straits
except Timor Passage. Comparisons between observations and reanalysis data taken only at the mooring locations showed that
not considering all grid points can improve the agreement between observational and reanalysis-based transport values in some
straits (Makassar and Ombai), possibly indicating observational undersampling in these straits or biased structures of the flow
(in reanalyses) but can lead to strong over- and underestimation in other straits (Lombok and Timor) as a result of mislocated
maxima in the passages. Given the biased structure of the flow in some straits, the usefulness of reanalyses to choose suitable
mooring sites is limited. Despite the mean velocity biases evident in most reanalyses, the observations were within the range
of the GREP in most cases. This study shows that the complex structure of the ITF region remains a challenging area for
reanalyses, but higher resolving products (GLORYS12V1) provide a promising outlook.


In the second part of the paper, we explored the annual cycle and vertical structure of the flow and involved processes in
more detail. The mean annual cycle of ITF transport, which is strongly influenced by Kelvin waves in all passages, highlighted
the importance of representing Kelvin wave activity in the reanalyses. Reanalyses generally exhibit too weak flow in greater
depths and weaker Kelvin wave activity in deeper layers compared to expectations from observational studies like Sprintall
et al. (2009). We also addressed the apparent time lag of one month in the mean annual cycle between the observations and
all considered reanalysis products in Makassar Strait. The lag in the inflow was largely removed when taking into account
additional measurements/reanalysis data from Lifamatola Passage, which indicates that the seasonal distribution of flow be-
tween Makassar and Lifamatoloa Passage is different in reanalyses compared to observations. However, it must be noted that
observational coverage in Lifamatola Passage is suboptimal.


For a better understanding of the vertical structure of the flow, we investigated the relationship between the vertically-varying
horizontal pressure gradient and ASV anomalies in Makassar. We found that different mechanisms are responsible for driving
the upper (<300 m) and lower (>300 m)-layer flow: the sea level gradient regulates transport anomalies in the upper layer, and



baroclinic effects dominate in lower layers. Their origins are the subject of current research and have been mostly attributed
to baroclinic Kelvin waves that originate in the equatorial Indian Ocean (Pujiana et al., 2019). A more accurate representation
of deep-layer ASVs in reanalyses is necessary to investigate the two-layer mechanism in detail and the corresponding impact
of Kelvin waves on the ITF. We also found that the net ITF response to ENSO (typically weaker during El Niño and stronger
during La Niña) is the result of dominant signals in the upper layer and counteracting anomalies in the lower layer. The balance
of these opposing signals differs between observations and reanalyses (and within reanalyses). Accurate representation of these
processes in reanalyses is clearly needed because of their relevance for climate monitoring. However, long-term observations,
especially in the outflow passages, are also necessary to improve the observational sampling of interannual and longer-term
variations of the ITF.

*Data availability.* The INSTANT data sets can be accessed online (http://www.marine.csiro.au/%7Ecow074/index.htm). The employed
ocean reanalysis data can be obtained from the Copernicus Marine Data Store (https://data.marine.copernicus.eu/products). The NINO3.4 in-
dex is a product of the National Oceanic and Atmospheric Administration (NOAA) Physical Sciences Laboratory (PSL) using the HadISST1
dataset (https://psl.noaa.gov/gcos_wgsp/Timeseries/Data/nino34.long.data).



# Appendix A

## A1



**Figure A1.** Mean ASV cross sections for a-c) Lombok Strait, d-f) Ombai Strait, and g-i) Timor Passage as given by the observations (left), the GREP mean (middle). The far right column corresponds to the respective differences between observations and GREP mean with the RMSE given in the top right corner. White dash-dot lines represent mooring locations. Negative values indicate southward-directed velocities (towards the Indian Ocean).





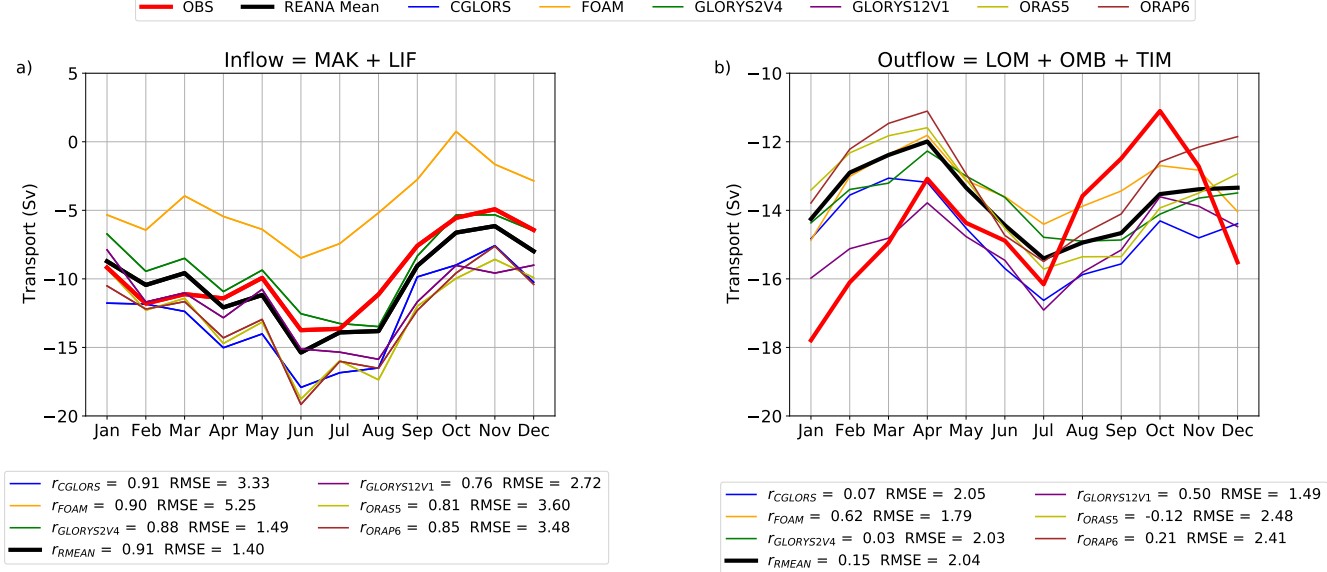

**Figure A2.** Mean annual cycles of ITF transport through a) the inflow passages (= Makassar Strait + Lifamatola Passage) and b) the outflow passages (= Lombok Strait + Ombai Strait + Timor Passage) as represented by observations and reanalyses. Observations in Lifamatola Passage are only available below 400 m. Subjacent boxes display corresponding Pearson correlation coefficients $r$ and RMSEs. Note that all annual cycles refer to the INSTANT period (January 2004-December 2006). Negative values indicate southward-directed transports (towards the Indian Ocean).





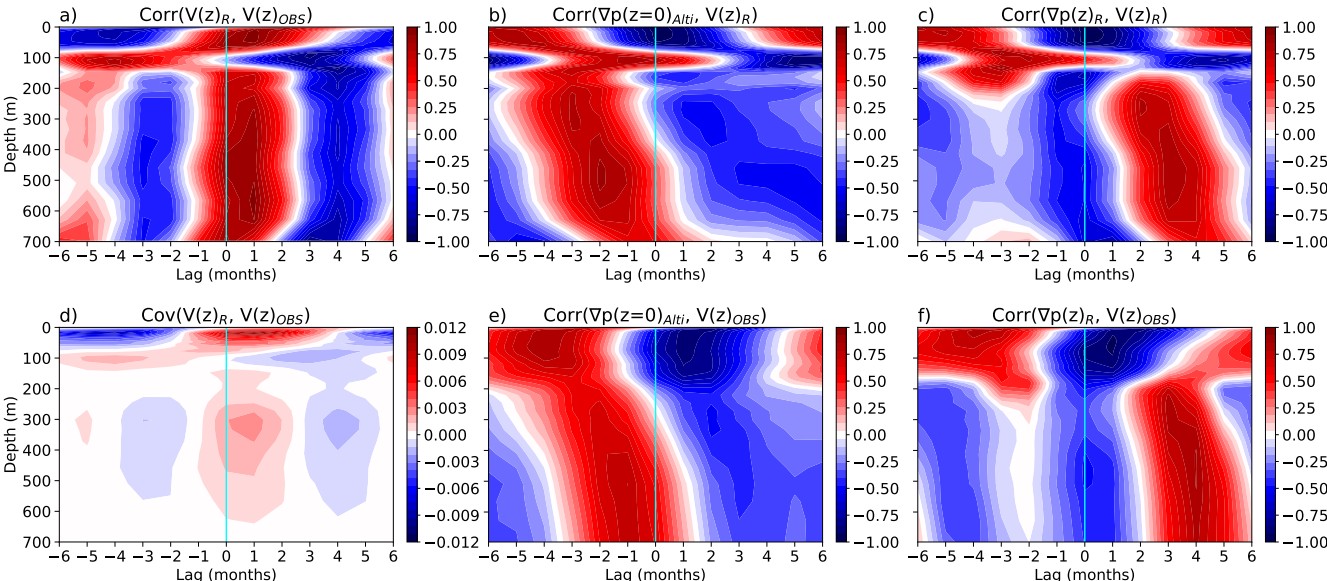

**Figure A3.** Analogous to Fig. 12 except the observed and reanalysis based monthly time series are replaced by 10-year time series consisting only of the mean annual cycle. Sea level gradients/pressure gradients refer to the ITF's entrance and exit region. Considered time series cover the period between 2004 and 2017. Red (blue) values indicate positive (negative) correlations.

*Author contributions.* Magdalena Fritz and Michael Mayer conceptualized the study. All co-authors were involved with data preparation and
analysis as well as interpretation of results. Magdalena Fritz prepared the manuscript with contributions from all co-authors.

*Competing interests.* The authors declare that they have no known competing financial interests or personal relationships that could have
appeared to influence the work reported in this paper.

*Acknowledgements.* We sincerely thank Rebecca Cowley from the Commonwealth Scientific and Industrial Research Organisation (CSIRO)
for clarifying any issues with the INSTANT data set and for cross-checking first attempts. We would also like to acknowledge the help of Janet
Sprintall from the Scripps Institution of Oceanography at the University of California San Diego, for reviewing our preprocessing methods
and providing additional material. We thank Michael McPhaden from the National Oceanic and Atmospheric Administration (NOAA) for
sharing his expertise and providing useful feedback on our diagnostics. Open access funding provided by University of Vienna. Magdalena
Fritz and Michael Mayer received funding from CMEMS 21003-COP-GLORAN Lot 7. Michael Mayer and Susanna Winkelbauer received
additional funding from Austrian Science Fund project P33177.



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
