# Peer review of "Assessment of Indonesian Throughflow transports from ocean reanalyses with mooring-based observations"

_EGUsphere, 2023_

## Author Response (AR1)

Legend: Comments are printed in *cursive font.* Response is printed in regular font (black). Adjustments in the manuscript are printed in regular font (red).

RC1

Overall comments:

*Assessment of Indonesian Throughflow transports from ocean reanalyses with mooring-based observations Magdalena Fritz, Michael Mayer, Leopold Haimberger and Susanna Winkelbauer. This paper is nicely written and organised. It compares observations collected in the ITF with output from several reanalysis products available via the Copernicus Marine Service and the mean product from them. The comparison with the observations shows that the finer 1/12° product better represents the ITF through nearly all of the passages, but the Timor passage is difficult to characterise in these reanalysis products. The discussion is clearly set out and valuable for those using the reanalysis products and developing new products, highlighting the need to characterise this difficult region of the ocean.*

Thanks for the appreciative comment.

Major comments:

*The paper could be published with just a few minor adjustments/additions as listed below.*

Minor comments:

*Comment: Figure 1 refers to dashed red arrows but there are only solid in the figure. Suggest using another colour for the black boxed entrance and exit areas (perhaps green) to stand out more.*

Response: Reference to dashed red arrows has been removed. Green boxes have been added.

*Comment: Line 67: Add Gordon et al 2019 reference to MITF campaign*

Response: Done.

*Comment: Line 86: Can you briefly describe the special treatment required for mooring blow-over, or provide a reference?*

Response: Measurements from ADCPs require special treatment to account for mooring blow-over. We determine the actual measuring depths by subtracting the ADCP's range of bins from its location at each time stamp, as given by the pressure time series. We choose the lowest measured pressure value and assume it to be the mooring's depth at rest. We added this additional explanation.

*Comment: Line 127: replace 'moorings' with 'profiling floats'*

Response: Done.

*Comment: Figure 4: Panel label on b) perhaps should use 'GREP' rather than 'REANA' (same with all other figures)? In the caption, suggest putting '1/4° with the b) descriptor, remove the 'upper panel' and 'lower panel' wording.*

Response: Figure titles and text descriptions should indeed be consistent and we made the appropriate changes. We kept the 'upper' and 'lower' panel wording in order to avoid any confusion.

*Comment: Line 261: Is using boreal summer/winter references appropriate given that Indonesia has two seasons (dry/rainy or monsoon/dry etc)? There are 'winter' and 'summer' references elsewhere, maybe consider local seasonal references throughout the paper.*

Response: The wording was adapted from Pujiana et al., 2019. Boreal winter/summer are used to standardize which season we are referring to and to group the northwest and southeast monsoon in its seasons. Furthermore, we are under the impression that boreal summer/winter coincides with wet/dry season. Therefore, we kept the boreal summer/winter wording.

*Comment: Figure 7 caption: suggest: "upper legend displays correlation coefficients between transport anomalies and observation anomalies r at lag = 0'*

Response: Done.

*Comment: Line 493: hyperlink to INSTANT data is correct, but the text has some odd characters in it ('%7E' instead of '~').*

Response: Done.

RC2

*This paper assesses 6 ocean reanalyses products with mooring data from various channels that measure the Indonesian Throughflow (ITF). While there are really no new scientific results to come out of this paper, the manuscript does a great job of showing the strength and weaknesses of these products that can be challenged in the narrow passages and swift flows of the ITF. In particular, the paper compares 5 reanalyses products that are ¼ degree with a newly available higher resolution 1/12 degree GLORYS12V1 product. The main purpose here seems to be a rigorous assessment of how well the reanalyses perform in the region so as to be able to undertake scientific analyses in future papers. With that motivation in mind, I think the paper is very well written and have only a few relatively minor suggestions that the authors might consider before the editor accepts the manuscript for publication.*

Thanks for appreciating the main aim of the manuscript. While we agree that the paper does not shift paradigms about the processes going on in the ITF region, we consider the quantification of uncertainties in the used data sets, and in particular, the found marked differences in the strength of the pressure gradient signal in greater depths, to which the reviewer is referring to in his/her next comment, as progress beyond the state of the art in this area.

*In general, I find it a bit problematic that the reanalysis products (and the underlying NEMO model?) do not resolve the linear wave dynamics of Kelvin waves that are critical to the temporal variability of the ITF. I find that puzzling and warrants further analysis by the group and further discussion in the manuscript.*

Response: As we point out several times throughout the paper, e.g., when discussing the annual cycle of the ITF (description of Fig. 5+6), there is definitely Kelvin wave activity represented in the reanalyses, but due to their too weak representation of flow in greater depths, Kelvin wave signals are considerably supressed. However, they also influence the flow near the surface, and this seems to be represented in the reanalyses. We phrased this more explicitly in the description of Fig. 5 and Fig. 6.

*Comment: Line 150: I am unclear what correlation is performed here. It seems to be some mixture of depth and perhaps time? The authors need to more clearly state how the correlations are calculated and what they are between.*

Response: We revised the description of Fig. 3: Purely spatial pearson correlation coefficients between observation-based and reanalyses-based temporally averaged ASV profiles across depth are given in the boxes in the lower left. This is now also mentioned in the text.

*Comment: Figure 3 only gives the vertical ASV profile of the products and observations in Makassar Strait. Later in the paper, in Figure 9 the manuscript reports on the rmse differences for the ASV vertical profiles in the exit passages of the ITF, although these are never shown. For complete transparency, it would be very useful to show the vertical profiles of ALL passages, and this could be shown in the Appendix.*

Response: ASV profiles for all mooring locations are available. We added a figure (Fig. A1) with all additional profiles (Lombok, Ombai, and Timor) to the Appendix.

*Comment: Figure 4: The colorbar used for the velocity here is hard to distinguish. Could a more differential color bar be used so that it was clearer where the maxima and minima occur? Indeed, the choice of red and blue can also be problematic for those readers who might be color blind.*

Response: We suggest a different colormap for the cross sections of ASV which is now used in Fig. 4, Fig. 5, and Fig. A2. However, we would like to keep the red and blue colormap for difference- and anomaly plots since keeping values close to zero white can be visually helpful.

*Comment: Line 172. The cross-strait differences in the strength of the flow through the outflow passages was also noted in Sprintall et al (2009) and that should be cited here.*

Response: Thanks for pointing this out. We added the reference.

*Comment: Line 187 and following discussion. The mean integrated transport results appear very sensitive to the bathymetries. What bathymetry does NEMO use? How well are the reported sill depths resolved by the model? The sentence on line 194-196 seems to blame the lack of agreement between model and observed transports on the observational uncertainties caused by the extrapolation of two measured profiles. However I think the observations are on much stronger ground than the model here, which probably only has 2-3 grid points in the horizontal direction to resolve the flow. The sentence needs to be modified to reflect that.*

Response:

The NEMO documentation provides the following information: The bathymetry is usually built by interpolating a standard bathymetry product (e.g., ETOPO2) onto the horizontal ocean mesh [1]. We conclude the bathymetry in the employed reanalysis products is realistic, but representation of small-scale features and exact sill depths is likely limited by the resolution of the products.

We agree with the reviewer that the ¼ degree products have only very few grid points in the narrow straits. However, we would like to emphasize that the main aim of the concerning paragraph is to point out the sensitivity of the resulting integrated flow to the used bathymetry and exact extrapolation method – no matter whether one uses observational or reanalysis-based flow data. Small differences in the method of extrapolating only two measured profiles can lead to deviations in mean integrated transport values. The mean integrated transports are indeed very sensitive to how the bathymetry is used for the cross-passage extrapolation of ASVs and also the use of the boundary conditions. The sentence on line 194-196 refers to the differences between our observation-based results and the observations-based results from Gordon et al. (2008). We made small modifications to the text to make the message clearer: We view the implemented bathymetries in the employed products as realistic, but representation of small-scale features and precise sill depths is likely limited by the resolution of the reanalyses.

*Comment: Figure 6: Again, I am confused as to what correlation coefficient is reported here. Is it the correlation between observations and the products over the 12 monthly data points? Can this be explicitly said? I struggle to understand why the Timor Passage correlations are seemingly very high,*

*especially given that none of the products look at all like the observations? This needs further discussion.*

Response: Modified figure description: Legends display correlation coefficients r and RMSEs between observations and the respective reanalysis over the 12 monthly data points.

Since there are only 12 data points per time series to perform the correlation with, the results should generally be interpreted with caution. This is, e.g., clearly visible in the difference between ORAS5 and ORAP6: while their transport curves are similar but opposite to the observed transport (hence the negative correlation), they exhibit very different correlation coefficients. We consider removing he correlation coefficients.

*Comment: Figure 7: Somewhat surprisingly, the GLORYS12V1 product seems to have a relatively poor correspondence of the temporal variability and also the correlation with the ENSO index compared to some of the GREP products. This probably warrants further discussion in the manuscript.*

Response: Indeed, the temporal variability of GLORYS12V1 is not strongly pronounced in Fig. 7, as also reflected by the lower STDDEV discussed in the text. In addition, the mean annual cycle as represented by GLORYS12V1 is also weakest in the other straits, apart from Lombok Strait. These aspects are now said more explicitly in the text.

*Comment: Line 355: This paragraph seems to be an assessment of the reanalyses products with the currents observed at the individual mooring sites in each strait. But this comparison does not appear to be actually shown in the manuscript – there appears to be no figure that actually shows that comparison? (Figure 9 appears to be the time averaged vertical profile integrated across each strait?). I suggest deleting this paragraph or showing the results.*

Response: The results from Fig. 9 are derived from the mean ASV profiles in each strait, therefore showing ASV profiles from all straits as suggested earlier might clarify this paragraph. We added the figure to the Appendix (Fig. A1).

*Comment: Line 357: By "opposite transport orientation in the upper ~200 m" do you mean the reversal in transport direction between the upper 200 m and that below? If so, then can you reword this to be clearer please?*

Response: Revised sentence: Also, in Ombai Strait, opposite transport orientation between the north and south in the upper ~200 m is only represented in GLORYS12V1.

*Comment: I don't think that subsequently in the abstract and conclusions (for example, the discussion in the paragraph beginning line 454) enough credit is given to the CGLORS product that seems to do a much better job of representing the seasonal cycle that GLORYS12V1, at least in terms of the performance summary metrics.*

Response: True. Added sentence in conclusion: In terms of representing the annual mean bias and the mean seasonal cycle (Fig. 10a and 10b, respectively), CGLORS shows generally high skill, especially in Ombai Strait and Timor Passage (even outperforming the higher-resolving GLORYS12V1).

*Comment: Line 395: The reported correlations between the upper and lower layer flow in Makassar Strait are quite low. Are they significant?*

Response: Yes. (pvalue_OBS = 0.0023, pvalue_GREP = 0.0104, pvalue_GLORYS12 = 7.7e-5). Added p-values and the following comment to the text. For the calculation of the respective p-values we considered the autocorrelation of the time series and estimated n_eff based on equation B12 in [2].

[1] Madec, G., and the NEMO team, 2008: NEMO ocean engine. Note du Pôle de modélisation, Institut Pierre-Simon Laplace (IPSL), France, No 27, ISSN No 1288-1619.

[2] Oort, A., and Yienger, J., 1996: Observed Interannual Variability in the Hadley Circulation and Its Connection to ENSO. Journal of Climate, Vol. 9, 2751-2767. Doi: https://doi.org/10.1175/1520-0442(1996)009%3C2751:OIVITH%3E2.0.CO;2